# REWEIGHTOOD: LOSS REWEIGHTING FOR DISTANCE-BASED OOD DETECTION

## ABSTRACT

Out-of-Distribution (OOD) detection is crucial for ensuring the safety and reliability of neural networks in critical applications. Distance-based OOD detection is based on the assumption that OOD samples are mapped far from In-Distribution (ID) clusters in embedding space. A recent approach for obtaining OOD-detection-friendly embedding space has been contrastive optimization of pulling similar pairs and pushing apart dissimilar pairs. It assigns equal significance to all similarity instances with the implicit objective of maximizing the mean proximity between samples with their corresponding hypothetical class centroids. However, the emphasis should be directed towards reducing the Minimum Enclosing Sphere (MES) for each class and achieving higher inter-class dispersion to effectively mitigate the potential for ID-OOD overlap. Optimizing low-signal dissimilar pairs might potentially act against achieving maximal inter-class dispersion while less-optimized similar pairs prevent achieving smaller MES. Based on this, we propose a reweighting scheme **ReweightOOD**, that adopts the similarity optimization which prioritizes the optimization of less-optimized contrasting pairs while assigning lower importance to already well-optimized contrasting pairs. Such a reweighting scheme serves to minimize the MES for each class while achieving maximal inter-class dispersion. Experimental results on a challenging CIFAR100 benchmark using ResNet-18 network demonstrate that the proposed reweighting scheme improves the FPR metric by a whopping 38% in comparison to the baseline. In various classification datasets, our method outperforms existing methods, making it a promising solution for enhancing OOD detection capabilities in neural networks.

## 1 INTRODUCTION

OOD detection refers to detecting the samples lying beyond the scope of training distribution. During the inference phase, it is indeed imperative to prevent the prediction of unknown samples, referred to as OOD samples, as the model lacks familiarity with such instances, and consequently, they should be accurately flagged. This issue becomes even more critical in domains like autonomous driving and medical imaging, where entrusting neural networks to handling unforeseen scenarios is detrimental. In these contexts, either relinquishing appropriate control to human discretion or flagging the instance becomes essential. The incorporation of OOD detection mechanisms holds paramount importance in ensuring safety and reliability. Rather than solely excelling at the primary task, models are now expected to possess the capability of identifying OOD samples effectively too.

OOD samples inherently possess distinct characteristics that set them apart from in-distribution (ID) data. These differentiating characteristics can be observed in softmax probability (Hendrycks & Gimpel, 2017), embedding space (Sun et al., 2022; Lee et al., 2018), or in some scoring functions (Liu et al., 2020; Wang et al., 2022). Distance-based methods exploit the embedding space to quantify the OOD-ness of the samples. Two popular postprocessing approaches in distance-based OOD methods are Mahalanobis distance (Lee et al., 2018) and K-nearest neighbor (Sun et al., 2022). A key assumption of these approaches is that OOD samples lie far away from ID clusters. Hence, the focus should be on obtaining such desirable embedding space for superior OOD detection performance.

Training-time regularization techniques can be employed to regularize neural networks to enhance OOD detection. Some approaches (Wei et al., 2022; Regmi et al., 2023) utilize the hyperspherical constraint during cross-entropy-based training to reduce overconfidence. Contrastive learning, a recent alternative, which also deals with hyperspherical embeddings to promote class-separable representations, as demonstrated by self-supervised contrastive learning (Sehwag et al., 2021), supervised contrastive learning (SupCon) (Khosla et al., 2020), and CIDER (Ming et al., 2023). While these contrastive-based methods, when coupled with distance-based postprocessing, show promise, they lack consideration for the current proximity of the contrasting pairs during training. In essence, they consistently optimize the cosine similarity without considering whether the pairs have been adequately optimized. Our proposition suggests reweighting contrastive pairs based on cosine similarity in the embedding space betters OOD performance. Specifically, we propose prioritizing pair instances where their corresponding embeddings are not aligned, and deprioritizing pair instances that are sufficiently aligned. By dynamically adjusting loss weights based on embedding space proximity, contrastive learning can focus more on challenging or unoptimized pairs, thereby reducing the Minimum Enclosing Sphere (MES) for each class and maximizing inter-class dispersion.

Hence, we present an effective OOD detection framework **ReweightOOD** based on loss reweighting. Our reweighting mechanism consists of a linear transformation of the cosine similarity followed by the application of the reweighting function. We employ scaling and shifting operations to achieve the desired range, and we employ the sigmoid function as reweighting function. This approach improves the OOD detection by a significant 38% improvement in FPR metric in a challenging CIFAR100 benchmark using ResNet-18 network. Our approach outperforms the current approaches making it a promising approach for detecting OOD samples. We summarize our contributions in the following points:

- We propose a similarity reweighting framework **ReweightOOD** in contrastive optimization for superior distance-based OOD detection. We show a simple reweighting mechanism can improve the performance by **38%** in the FPR metric in challenging the CIFAR100 benchmark.

- We provide the design of the reweighting mechanism for the first time in OOD detection by coupling linear transformation and sigmoid weighting function. We illustrate that our domain for the reweighting mechanism can be flexibly adjusted by scaling and shifting using hyperparameters.

- We reveal the implication of the reweighting in achieving an MES of a smaller radius for all classes and higher class-centroid dispersion in the embedding space. Specifically, in a challenging CIFAR100 benchmark, the reweighting mechanism **reduces mean MES radius by 14.28%** and **increases mean inter-class dispersion** by a factor of $\sim 2$.

## 2 Preliminaries

### 2.1 Out-of-Distribution Detection

We consider multi-class classification scenario, wherein $\mathcal{P}_{\text{in}} = (x_i, y_i)_{i=1}^{N}$ represents the training distribution, commonly referred to as the In-Distribution. In this context, the tuple $(x_i, y_i)$ signifies an image-label pair, where $y_i$ is an element of the set $\{1, 2, \ldots, C\}$, with $C$ representing total number of classes. During the testing phase, samples from a distribution $\mathcal{P}_{\text{out}}$, differing from the training distribution $\mathcal{P}_{\text{in}}$, are encountered. Out-of-distribution (OOD) detection is framed as a binary classification task, where a scoring function $SC(\mathbf{x})$ and a threshold $\lambda$ guide the decision process with those exceeding $\lambda$ labeled as ID and the rest as OOD. The threshold $\lambda$ is often set for a 95% true positive rate on training data.

### 2.2 Hyperspherical Embeddings

The embeddings lying on the surface of the hypersphere of radius $r_h$ are known as hyperspherical embeddings. An embedding can be transformed into a hyperspherical one by employing $L_2$ normalization. We employ our reweighting mechanism in contrastive training after transforming the raw embedding into the hyperspherical one.

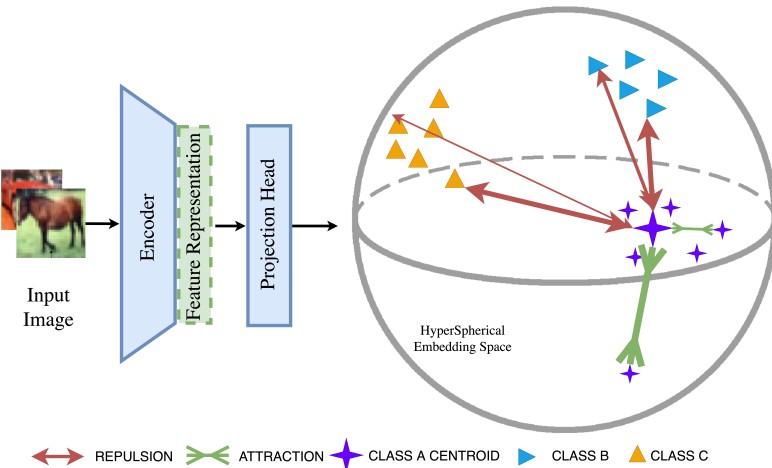

Figure 1: The figure shows an overview of the optimization process in ReweightOOD. ReweightOOD uses encoder (backbone) and the projection head to generate raw embeddings, which are subsequently converted into hyperspherical embeddings. The thickness of the depicted lines in the hypersphere visually represents the strength of the reweighting factor during pair optimization.

## 3 METHOD

### 3.1 OVERVIEW

An overview of the proposed OOD detection framework ReweightOOD is shown in Figure 1. ReweightOOD consists of backbone (encoder) network $f_\theta$ and projection head $g_\theta$. The hyperspherical representation $\hat{h}_i = g_\theta(f_\theta(x_i))$ is obtained from ReweightOOD framework for each image $x_i$. Hyperspherical representations $\{\hat{h}_i\}_{i=1}^N$ form contrastive pairs that are weighted on the basis of their respective cosine similarities prior to contrastive optimization.

### 3.2 CONTRASTIVE OPTIMIZATION

Contrastive learning aims to learn useful representations by maximizing within-class cosine similarity $s_w$ and minimizing between-class cosine similarity $s_b$. If $\left(x_{anchor}^j, x_{pos}^j\right)$ and $\left(x_{anchor}^i, x_{neg}^i\right)$ are pairs of images of the same class and different classes respectively, any given instance of within-class cosine similarity $s_w^j$ and between-class cosine similarity $s_b^i$ can be expressed as $s_w^j = h_{anchor}^j \cdot \hat{h}_{pos}^j$ and $s_b^i = h_{anchor}^i \cdot \hat{h}_{neg}^i$ where $h$ denotes latent representation of respective inputs. Considering the availability of $n$ between-class similarity $s_b$ and $o$ within-class similarity $s_w$, loss formulation for a sample $k$ in a batch of size $(n + o + 1)$ with temperature $\tau$ can be expressed as:

$$\mathcal{L}_k = \log\left(\sum_{i=1}^n \exp(s_b^i/\tau)\right) - \log\left(\sum_{j=1}^o \exp(s_w^j/\tau)\right) \tag{1}$$

**Assumption of 1** This (unweighted) optimization assumes an equal role of each between-class similarity $s_b$ and within-class similarity $s_w$ in obtaining optimal embedding space for distance-based OOD detection. In the subsequent section, we delve into the implication of this assumption.

### 3.3 IMPLICATION OF UNWEIGHTED OPTIMIZATION

Contrastive learning attempts to map all the instances of a category to its ideal centroid. However, as shown schematically in Figure 2 (a), the complexities inherent in real-world images make the ide-

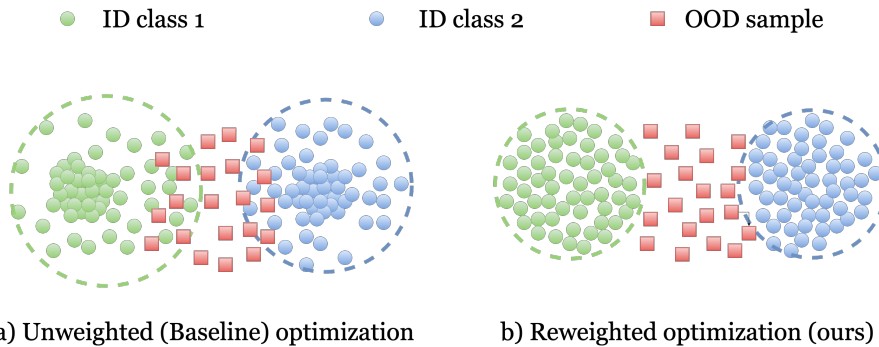

Figure 2: Comparison of (a) Unweighted optimization and (b) Reweighted optimization leading to different extents of overlapping with OOD samples in the embedding space.

alistic goal of mapping all instances of a class very close to its ideal centroid in contrastive learning impractical. We define samples that are easy to pull near the centroid as *easy positives* and those samples that are difficult to pull near the centroid as *hard positives*. The compact clustering of *easy positives* around the centroid, as shown in Figure 2 (a), adds practically no value in OOD separation. However, as shown in Figure 2 (b), trading off the easy compact clustering with more weightage given on optimizing (pulling) hard positives around the centroid has a potentially beneficial effect on obtaining the Minimum Enclosing Sphere of smaller radius for all classes. Obtaining a smaller MES radius has a direct advantage linked with a smaller possibility of ID-OOD overlapping, as shown in Figure 2 (b).

Furthermore, given an anchor sample, samples that are easily distinguishable from the anchor can be referred to as *easy-negatives*. Conversely, samples that are similar and not easily distinguishable from the anchor sample can be referred to as *hard negatives*. In a multi-class setup, there is a greater presence of *easy-negatives* that don't provide useful learning signals. Optimizing these *easy negatives* can rather be a noisy process that potentially hinders the maximal inter-class dispersion. Furthermore, *hard negatives* are more informative for maximizing inter-class dispersion. From a separate perspective, *hard negatives* have a greater likelihood of getting overlapped with OOD instances. Hence, suppressing the effect of *easy negatives* and prioritizing *hard negatives* seem to be of utmost importance for maximizing the inter-class dispersion.

### 3.4 REWEIGHTING MECHANISM FOR SIMILARITY SCORES

Hence, a requirement for the optimum embedding learning for OOD detection is: to give more importance to samples that are difficult to align (*hard negatives* and *hard positives*). Since similarity during optimization can convey the difficulty of the sample, the requirement for designing the reweighting mechanism is to make it the function of the similarity score. We use the linear transformation of the score and apply the sigmoid function to obtain the reweighting factor. Linear transformation basically consists of two operations:

**Scaling** The original range of cosine similarity is $[-1, 1]$. The scaling operation is utilized to rescale the similarity scores prior to the weighting function. Specifically, the scaling enables adjustment of the slope of the weighting function, thereby controlling the rate of increase in the reweighting factors based on the similarity scores. Scaling similarity scores $s$ with scalar $m$ resulting in the domain $[-m, m]$ from $[-1, 1]$, the weighting factor can be given as: $\mathcal{S} \rightarrow s \cdot m$

**Shifting** Shifting allows shifting of the domain of rescaled similarity $\mathcal{S}$ by given scalar $c$ to determine the desirable part of the weighting function depending on the nature of similarity score $s$. $\mathcal{T} \rightarrow \mathcal{S} + c, \mathcal{T} \rightarrow s \cdot m + c$

**Final Linear Transformation** Hence, the final linear transformation can be expressed as $\mathcal{T} = s \cdot m + c$. As the two similarity scores might have different optimal hyperparameters, we allow defining different sets of linear transformation. So, we denote two such linear transformations as $\mathcal{T}_{\mathcal{B}} = s_b \cdot m_b + c_b$ and $\mathcal{T}_{\mathcal{W}} = s_w \cdot m_w + c_w$ for between-class and within-class similarities.

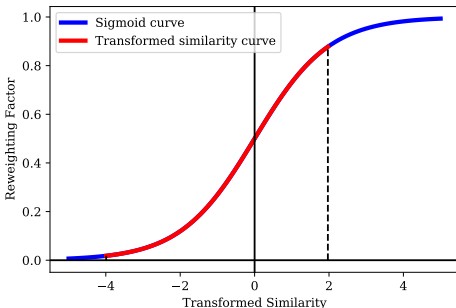 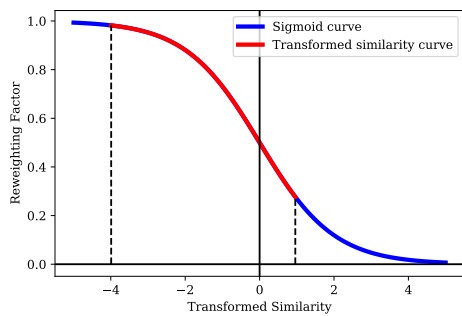

Figure 3: Reweighting mechanism for $s_b$      Figure 4: Reweighting mechanism for $s_w$

**Reweighting function** The scaled similarity scores are then passed through the weighting function, which maps them to values in a predefined range to obtain reweighting factors. Since we already established in the previous section regarding the unequal role of various similarity scores in obtaining optimal embedding for OOD detection, we need to impose the increasing importance of between-class similarity $s_b$ as it progresses towards positive value from negative value. Since the range of similarity scores due to linear transformation can be both negative as well as non-negative, we propose sigmoid function for obtaining its reweighting factor as shown in Figure 3. The sigmoid function can be expressed as : $\sigma(\mathcal{T}_\mathcal{B}) = \frac{1}{1+e^{-\mathcal{T}_\mathcal{B}}} = \frac{1}{1+e^{-s_b \cdot m_b - c_b}}$

Similarly, we have already established the decreasing importance of within-class $s_w$ similarity scores as it progress towards positive value from negative value, we need the reweighting function for $s_w$ to possess such characteristics. Hence, we propose *reverse-sigmoid* function for $s_w$ reweighting as shown in Figure 4. It is basically the modified version of the sigmoid function which can be expressed as : $\sigma'(\mathcal{T}_\mathcal{W}) = \frac{1}{1+e^{\mathcal{T}_\mathcal{W}}} = \frac{1}{1+e^{s_w \cdot m_w + c_w}}$

So, accommodating the reweighting mechanism in 1, the optimization then can be reformulated as:

$$\mathcal{L} = \log\left(\sum_{i=1}^{n} \exp(\sigma(\mathcal{T}_\mathcal{B}^i) \cdot s_b^i/\tau)\right) - \log\left(\sum_{j=1}^{o} \exp(\sigma'(\mathcal{T}_\mathcal{W}^j) \cdot s_w^j/\tau)\right) \quad (2)$$

$$= \log\left(\sum_{i=1}^{n} \exp(\frac{1}{1 + e^{-s_b^i \cdot m_b - c_b}} \cdot s_b^i/\tau)\right) - \log\left(\sum_{j=1}^{o} \exp(\frac{1}{1 + e^{s_w^j \cdot m_w + c_w}} \cdot s_w^j/\tau)\right) \quad (3)$$

**Reweighting flexibility and bounded range** The transformation consisting of scaling and shifting allows flexible control over reweighting specific to the nature of similarity (between-class and within-class). This allows a better bet in obtaining optimal embedding for OOD detection. Furthermore, the sigmoid function exhibits a bounded range that lies within the interval [0, 1]. The bounded range makes the weighting mechanism controlled and stable.

## 4 EXPERIMENTS

**Datasets** The ID datasets CIFAR10 and CIFAR100 (Krizhevsky et al., 2009) are used for the training models from scratch while ImageNet100 is used for fine-tuning pretrained models. The OOD detection performance of CIFAR datasets is evaluated in the following datasets: MNIST (Deng, 2012), iSUN Xu et al. (2015), LSUN-r (Yu et al., 2015), LSUN-c (Yu et al., 2015), SVHN (Netzer et al., 2011), Textures (Kylberg, 2011), and Places365 (Zhou et al., 2017). For ImageNet100, the OOD datasets used are iNaturalist (Van Horn et al., 2018), SUN (Xiao et al., 2010), Places365 (Zhou et al., 2017), and Textures (Kylberg, 2011), NINCO (Bitterwolf et al., 2023), OpenImage-O (Wang et al., 2022), and Semantic Shift Benchmark (SSB) (Vaze et al., 2021).

**Metrics** We mainly use two OOD metrics (AUROC and FPR@95) to quantify the OOD detection performance. AUROC stands for the Area Under Receiver-Operator Characteristics, and FPR@95

stands for False Positive Rate @ 95. A higher AUROC score quantifies a higher probability of correct OOD/ID classification, and a lower FPR suggests a lower probability of ID samples getting misclassified as OOD.

**Training pipelines** Similar to previous approaches KNN+ (Sun et al., 2022) and CIDER (Ming et al., 2023), we perform experiments with non-contrastive approaches for 100 epochs and contrastive approaches for 500 epochs. For posthoc methods, we train a standard model using vanilla cross-entropy loss. We train our model with a learning rate of 0.5 using a cosine annealing decay schedule with a batch size of 512 and 0.0001 weight decay. The temperature parameter $\tau$ is set to 0.1. The hyperparameters are optimized with respect to the validation set (Gaussian noise). For a fair comparison, we train all the methods in the same setting. We use ResNet-18 architecture for CIFAR-10/100 experiments. The linear transformation hyperparameters $(m_b, c_b, m_w, c_w)$ for CIFAR100 and CIFAR10 experiments using ResNet18 network are set to $(5, -2, 2, 1)$ and $(5, -4, 2, 1)$ respectively. We also use WideResNet and DenseNet architecture to test architectural compatibility. The ablation regarding the linear transformation of the reweighting mechanism is provided in the appendix.

**OOD detection scores** Since we focus on learning a suitable embedding for OOD detection, we use two distance-based OOD scores in the embedding space: KNN and Mahalanobis distance. We use the KNN postprocessor by default and also investigate the performance with Mahalanobis distance (MDS). Like the previous approach CIDER, we use K=100 for CIFAR-10 experiments and K=300 for CIFAR-100 experiments for KNN postprocessor.

### 4.1 QUANTIFICATION OF EMBEDDING QUALITY FOR OOD DETECTION

**Minimum Enclosing Sphere** The concept of radius of the Minimum Enclosing Sphere (MES) serves to characterize the overall radius of a class while downplaying the significance of achieving a compact representation for easily distinguishable positive instances. This emphasis on class radius $r_{cl}$ is motivated by the observation that embeddings associated with samples far from its ideal centroid tend to intersect with OOD samples, thereby compromising the performance of OOD detection. In essence, from the lens of ideal perspective, the ID samples residing at the periphery of a class should ideally be closer to the empirical centroid to avoid compromise in OOD detection performance. This notion can be effectively encapsulated through the concept of MES radius. Thus, from the perspective of ID-OOD separability, the MES radius emerges as a suitable metric for quantifying the effective compactness of class embeddings. It follows that the smaller the MES radius, the greater the effective compactness of class $cl$ for OOD detection purposes. The empirical centroid, denoted as $\boldsymbol{\mu}_{cl}$, is a straightforward computation involving the summation of all embeddings corresponding to category $cl$ over the entire set of samples, given by $\boldsymbol{\mu}_{cl} = \frac{\sum_{\mathbf{h}_i \in \mathcal{H}_{cl}} \mathbf{h}_i}{N_{cl}}$ where $\mathcal{H}_{cl}$ denotes normalized embedding representation of all samples in category $cl$ and $N_{cl}$ is the total number of samples in category $cl$.

$$r_{cl} = \max_{\mathbf{h}_i \in \mathcal{H}_{cl}} |\mathbf{h}_i - \boldsymbol{\mu}_{cl}|_2 \, (\downarrow) \tag{4}$$

Table 1: MES radius for first 10 classes and mean over 100 classes of CIFAR100 datasets in unweighted and weighted optimization.

| Method | Apples | Aquarium Fish | Baby | Bear | Beaver | Bed | Bee | Beetle | Bicycle | Bottles | ... | Mean |
|---|---|---|---|---|---|---|---|---|---|---|---|---|
| Baseline | 1.09 | 1.11 | 1.07 | 0.93 | 0.97 | 1.07 | 1.03 | 1.07 | 1.10 | 1.07 | ... | 1.05 |
| SupCon | 0.97 | 1.01 | 0.97 | 1.00 | 1.00 | 0.98 | 0.99 | 0.97 | 1.04 | 1.03 | ... | 1.01 |
| **(ReweightOOD)** *Ours* | 0.95 | 0.90 | 0.90 | 0.89 | 0.84 | 0.90 | 0.89 | 0.91 | 0.99 | 0.96 | ... | **0.90** |

**Centroid Dispersion** To enhance OOD detection performance, it is essential to ensure that centroids are distributed sufficiently far apart, allowing for the effective delineation of OOD samples within the unoccupied space between these centroids. This notion of centroid dispersion can be precisely quantified by measuring the angular distance between the empirical centroids of two distinct categories. Mathematically, we represent the centroid dispersion between two categories, denoted as $cl_a$ and $cl_b$, as follows:

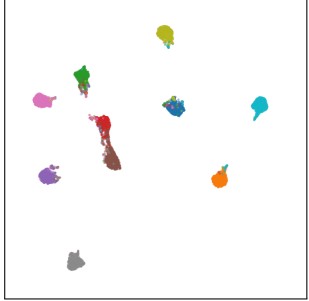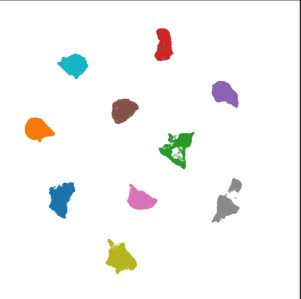

Figure 5: UMAP (McInnes et al., 2018) visualization of embedding space (CIFAR10) obtained from (left) CE objective and (right) ReweightOOD objective. The ReweightOOD objective allows the embeddings to be uniformly distributed and highly separable without class overlapping.

$$d_{ab} = \frac{\mu_a \cdot \mu_b}{|\mu_a|_2 \cdot |\mu_b|_2}, \quad a \neq b \, (\uparrow) \tag{5}$$

Overall embedding quality can be assessed by computing the mean dispersion value across all category pairs and the mean MES radius across all categories.

Table 2: Average centroid dispersion over 100 classes in CIFAR100 datasets.

| Method | Mean dispersion ($\uparrow$) |
| --- | --- |
| Baseline | 0.29 |
| SupCon | 0.42 |
| **(ReweightOOD)** *Ours* | **0.63** |

**Embedding observations:** Table 1 and Table 2 present statistics on MES radius and centroid dispersion. ReweightOOD objective optimizes for a smaller overall radius across all CIFAR100 categories, as evidenced in Table 1 which reduces ID-OOD overlapping. Additionally, higher dispersion due to ReweightOOD indicates that ID classes are sufficiently spread apart, facilitating meaningful distance mapping for OOD samples. Consequently, reweighted optimization yields improved embedding quality. Furthermore, the qualitative UMAP visualization comparing the embedding space obtained with the CE objective and ReweightOOD objective is shown in Figure 5. It shows uniformly dispersed and highly separable embeddings without class-overlapping.

## 4.2 EMPIRICAL ANALYSIS

**Quantitative results** Quantitative results including the extensive comparisons of current approaches along with our approach are presented in Table 3. For all experiments in Table 3, ResNet-18 is trained with CIFAR-100 as the ID dataset. The OOD performance is shown in two metrics (FPR and AUROC) only. We compare our results with current contrastive approaches as well as non-contrastive approaches. Posthoc methods are applied to the classification model trained with vanilla cross-entropy. All the experiments assume the unavailability of OOD / outliers during the training time. Posthoc methods include MSP (Hendrycks & Gimpel, 2017), ODIN (Liang et al., 2017), Mahalanobis (Lee et al., 2017), DICE (Sun & Li, 2022), Activation Shaping (ASH) (Djurisic et al., 2023), React (Sun et al., 2021), GradNorm (Huang et al., 2021), RankFeat (Song et al., 2022) and Energy (Liu et al., 2020). Two non-contrastive training-time regularization approaches are GODIN (Hsu et al., 2020) and LogitNorm (Wei et al., 2022). We use default hyperparameters provided in the original work whenever required. In contrastive approaches, we compare our method with ProxyAnchor (Kim et al., 2020), CSI (Tack et al., 2020), SSD+ (Sehwag et al., 2021), KNN+ (Sun et al., 2022), and CIDER (Ming et al., 2023). Our approach leads to the best performance in both metrics. Furthermore, we present the OOD detection performance in CIFAR-10 experiments in the appendix which also shows our approach being highly performant in comparison to both contrastive as well as non-contrastive approaches.

Table 3: Mean OOD detection performance for CIFAR-100 (ID) with ResNet-18.

| Method | MNIST | | iSUN | | LSUN | | LSUN-r | | SVHN | | Texture | | Places365 | | Average | |
|---|---|---|---|---|---|---|---|---|---|---|---|---|---|---|---|---|
| | FPR↓ | AUROC↑ | FPR↓ | AUROC↑ | FPR↓ | AUROC↑ | FPR↓ | AUROC↑ | FPR↓ | AUROC↑ | FPR↓ | AUROC↑ | FPR↓ | AUROC↑ | FPR↓ | AUROC↑ |
| **Without Contrastive Learning** | | | | | | | | | | | | | | | | |
| MSP | 86.05 | 78.75 | 66.97 | 84.91 | 78.21 | 80.84 | 68.19 | 84.36 | 63.87 | 86.42 | 79.88 | 79.12 | 80.98 | 78.92 | 74.88 | 81.92 |
| ODIN | 73.64 | 84.61 | 40.07 | 92.54 | 72.70 | 85.75 | 42.38 | 92.23 | 74.80 | 84.45 | 72.22 | 81.63 | 81.05 | 79.06 | 65.27 | 85.75 |
| Mahalanobis | 81.91 | 77.22 | 95.23 | 59.99 | 95.45 | 56.16 | 95.14 | 61.16 | 92.47 | 64.96 | 75.55 | 73.95 | 92.84 | 62.89 | 89.90 | 65.19 |
| Energy | 88.57 | 79.05 | 63.27 | 88.05 | 78.04 | 84.63 | 63.95 | 87.56 | 59.09 | 89.84 | 78.94 | 80.68 | 83.58 | 79.02 | 73.63 | 84.12 |
| DICE | 81.96 | 79.46 | 67.60 | 87.04 | 67.11 | 86.97 | 70.25 | 86.00 | 60.45 | 89.78 | 76.01 | 80.17 | 83.76 | 78.76 | 72.45 | 84.03 |
| React | 88.29 | 78.55 | 63.22 | 87.85 | 77.73 | 84.89 | 63.89 | 87.23 | 58.06 | 90.16 | 78.35 | 81.75 | 83.47 | 79.17 | 73.29 | 84.23 |
| ASH | 78.75 | 81.14 | 70.04 | 81.92 | 77.26 | 83.31 | 70.72 | 80.54 | 57.94 | 88.22 | 76.10 | 81.14 | 82.55 | 77.53 | 73.34 | 81.97 |
| GradNorm | 86.54 | 63.98 | 68.82 | 78.28 | 71.14 | 84.48 | 70.75 | 75.27 | 60.92 | 83.80 | 77.96 | 69.68 | 84.79 | 69.29 | 74.42 | 74.97 |
| RankFeat | 95.62 | 61.39 | 87.99 | 74.29 | 95.73 | 67.85 | 88.66 | 73.81 | 79.82 | 80.68 | 91.63 | 66.11 | 91.17 | 66.41 | 90.09 | 70.08 |
| GODIN | 48.88 | 92.09 | 22.14 | 96.00 | 63.91 | 85.55 | 19.05 | 96.72 | 70.66 | 86.74 | 56.49 | 89.37 | 78.95 | 78.12 | 51.44 | 89.23 |
| LogitNorm | 51.65 | 90.28 | 92.84 | 69.00 | 15.52 | 97.23 | 92.68 | 70.77 | 73.71 | 84.15 | 86.85 | 71.27 | 77.98 | 80.77 | 70.18 | 80.50 |
| **With Contrastive Learning** | | | | | | | | | | | | | | | | |
| ProxyAnchor | 65.96 | 78.93 | 88.90 | 77.71 | 57.29 | 88.28 | 86.30 | 77.60 | 31.16 | 93.47 | 57.54 | 88.30 | 77.25 | 79.69 | 66.34 | 83.43 |
| CSI | 75.27 | 82.20 | 68.37 | 81.91 | 49.43 | 89.11 | 66.19 | 83.17 | 65.83 | 81.21 | 77.53 | 75.13 | 79.11 | 79.80 | 67.74 | 81.22 |
| SSD+ | 82.52 | 76.80 | 79.71 | 83.85 | 49.86 | 89.91 | 78.00 | 85.19 | 23.03 | 95.70 | 59.72 | 88.22 | 77.80 | 80.86 | 64.38 | 85.79 |
| KNN+ | 76.21 | 83.06 | 67.44 | 85.12 | 55.09 | 86.30 | 67.59 | 85.59 | 44.03 | 91.85 | 47.91 | 90.08 | 78.63 | 78.19 | 62.42 | 85.74 |
| CIDER | 63.24 | 85.64 | 73.78 | 77.96 | 26.51 | 93.37 | 75.98 | 78.03 | 17.58 | 96.33 | 34.15 | 92.34 | 78.56 | 73.04 | 52.83 | 85.24 |
| Baseline | 78.91 | 69.01 | 85.09 | 84.28 | 41.09 | 91.93 | 79.90 | 85.07 | 25.25 | 94.63 | 46.38 | 90.33 | 74.44 | 80.50 | 61.58 | 85.11 |
| **(ReweightOOD)** *Ours* | 19.24 | 96.86 | 57.56 | 87.54 | 19.59 | 96.86 | 56.31 | 88.23 | 8.39 | 98.31 | 28.72 | 94.11 | 78.70 | 76.01 | **38.36** | **90.91** |

**Compatibility with Mahalanobis distance (MDS)** In addition to the non-parametric method KNN, we also analyze the empirical quality of the embedding produced by various contrastive approaches by the use of Mahalanobis distance. As can be observed from 4, the superiority of the embedding quality produced by our method is evident from FPR/AUROC scores.

Table 4: Compatibility with MDS using CIFAR-100 (ID) dataset in terms of FPR using ResNet18.

| Method | MNIST | iSUN | LSUN | LSUN-r | SVHN | Texture | Places365 | Average **FPR** ↓ |
|---|---|---|---|---|---|---|---|---|
| ProxyAnchor | 75.48 | 88.94 | 52.78 | 87.62 | 7.69 | 58.21 | 74.85 | 63.65 |
| SSD+ | 82.52 | 79.71 | 49.86 | 78.00 | 23.03 | 59.72 | 77.80 | 64.38 |
| CIDER | 76.82 | 74.10 | 21.40 | 75.40 | 9.78 | 45.27 | 74.37 | 53.88 |
| Baseline | 80.68 | 88.95 | 21.31 | 85.63 | 5.83 | 40.51 | 66.83 | 55.68 |
| **(ReweightOOD)** *Ours* | 51.58 | 62.91 | 17.00 | 62.00 | 5.89 | 44.29 | 74.83 | **45.50** |

**Accuracy** While improving OOD detection performance, neural network-based OOD detectors ideally should not compromise in accuracy. Training linear classifier on frozen features obtained with WRN-40-2 pretrained with ReweightOOD objective, we obtained a 75.54% accuracy on CIFAR100, similar to the 74.96% accuracy from the cross-entropy objective, demonstrating the effectiveness of ReweightOOD in both OOD detection and category classification.

Table 5: OOD detection performance in large-scale experiments (ImageNet-100) in terms of FPR by fine-tuning pretrained ResNet50.

| Method | iNaturalist | SUN | Textures | Places | SSB Hard | Ninco | Openimage | Average **FPR** ↓ |
|---|---|---|---|---|---|---|---|---|
| Baseline | 3.07 | 2.39 | 4.57 | 5.47 | 35.39 | 29.15 | 7.05 | 12.44 |
| SupCon | 2.43 | 1.98 | 2.59 | 5.43 | 34.25 | 25.58 | 5.28 | 11.08 |
| **(ReweightOOD)** *Ours* | 2.18 | 1.97 | 2.73 | 5.29 | 32.00 | 24.63 | 5.06 | **10.55** |

**Compatibility with various backbones** In addition to ResNet-18, we experiment with diverse backbones, including WideResNet (WRN-40-2) and DenseNet architectures, to assess the adaptability of our method. As depicted in Table 6, in comparison to the baseline (unweighted formulation) and SupCon, our approach consistently leads to superior performance across various architectures in terms of all OOD metrics. Specifically, compared to the baseline, our approach leads to  10% and 20% performance improvement in WRN-40-2 and DenseNet architectures.

**Evaluation on large-scale ImageNet-100 dataset** In addition to conducting experiments on the CIFAR datasets, we assess the efficacy of our approach on the large-scale ImageNet-100 dataset within the context of fine-tuning pretrained models. We use the ImageNet-100 dataset, a subset of ImageNet, as the ID dataset for finetuning the pretrained ResNet-50 model. ImageNet-100 consists of images from 100 randomly sampled categories from the ImageNet dataset. The projection head is a non-linear MLP with a projection dimension of 128. The first three layers of ResNet50 are frozen and only the last layer along with the projection head is fine-tuned for 10 epochs with a learning rate of 0.01 and weight decay of 0.0001 using cosine annealing. The linear transformation

Table 6: Architecture compatibility of various methods with CIFAR100 (ID) datasets.

| Method | Architectures | | | |
| | WRN-40-2 | | DenseNet | |
| | FPR↓ | AUROC↑ | FPR↓ | AUROC↑ |
|---|---|---|---|---|
| Baseline | 53.55 | 87.07 | 39.03 | 91.11 |
| SupCon | 49.95 | 87.75 | 44.28 | 90.10 |
| **(ReweightOOD)** *Ours* | **47.94** | **88.45** | **31.36** | **92.21** |

hyperparameters $(m_b, c_b, m_w, c_w)$ are set to $(5, -4, 2, 1)$. The performance is evaluated with KNN postprocessing (K=300). We compare the OOD detection performance of our method with baseline and SupCon loss in terms of FPR and AUROC metrics as shown in Table 5. It depicts the superior performance of our approach in comparison to compared losses. It provides further empirical justification for producing superior embeddings for OOD detection.

## 5    RELATED WORKS

**OOD detection**    Posthoc approaches of OOD detection derive scores from pretrained models without any retraining. Some of these approaches deal directly with output space (Hendrycks & Gimpel, 2017; Liu et al., 2020; Ming et al., 2022b; Sun & Li, 2022; Djurisic et al., 2023) while recently more approaches have attempted to exploit the information from embedding space (Lee et al., 2018; Sastry & Oore, 2020; Tack et al., 2020; Zhou et al., 2021; Sehwag et al., 2021; Sun et al., 2022; Ming et al., 2022a; Du et al., 2022; Ming et al., 2023) for OOD detection. Wang et al. (2022) proposes dealing with both spaces. Furthermore, some works (Song et al., 2022; Sun et al., 2021; Zhu et al., 2022) also deal with feature activations. (Huang et al., 2021) showed the usefulness of gradient information in OOD detection. Guo et al. (2017) proposed temperature scaling to improve neural network calibration. Some works Wei et al. (2022); Regmi et al. (2023) also make use of normalization in logit/feature space to mitigate the overconfidence issue in neural networks. (DeVries & Taylor, 2018; Hendrycks et al., 2019; Hsu et al., 2020) propose various ways of regularizing neural networks during training to enhance the OOD detection performance.

**Deep Metric Learning**    A fundamental focus of deep metric learning is to learn highly discriminative features. Research areas such as face recognition and face verification have seen the introduction of many useful loss functions on hyperspherical embeddings (Wang et al., 2018; Deng et al., 2019; Liu et al., 2017; Wang et al., 2017) to satiate this objective. (Techapanurak et al., 2020) deals with cosine loss to achieve hyperparameter-free OOD detection.

**Contrastive Learning**    (Chopra et al., 2005; Schroff et al., 2015; Sohn, 2016) were the earliest works that explored the concept of contrastive loss. In recent years, contrastive learning has garnered significant attention in the domain of vision representational learning, encompassing both unsupervised and supervised paradigms (Chen et al., 2020a;b; He et al., 2020; Robinson et al., 2021; Khosla et al., 2020). While the majority of these approaches explicitly formulate positive and negative pairs, some recent works (Bardes et al., 2022; Chen & He, 2021; Grill et al., 2020). exclusively concentrate on positive pairs only. Few works (Khosla et al., 2020; Sehwag et al., 2021; Tack et al., 2020; Ming et al., 2023) have explored the use of off-the-shelf contrastive learning in the context of OOD detection. However, it is noteworthy that contrastive learning in OOD detection has remained relatively understudied. Our work deals with a similar line of harnessing contrastive approach in OOD detection.

## 6    CONCLUSIONS

In summary, this study introduces the **ReweightOOD** reweighting scheme, aimed at enhancing embedding quality to improve OOD detection performance. Our approach focuses on optimizing the cosine similarity of contrasting pairs by considering their current proximity, assigning higher priority to less-optimized pairs and lower priority to well-optimized ones. Experimental results across various classification datasets demonstrate non-trivial performance enhancements resulting from our approach. Furthermore, we reveal that our reweighting method reduces the Minimum

Enclosing Sphere radius for each class and increases inter-class dispersion, thereby enhancing the separation between ID and OOD samples in the embedding space.

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
