APPENDIX

## .1 PSEUDOCODE OF REWEIGHTOOD

The general working algorithm of ReweightOOD framework is given in Algorithm 1. Moreover, the loss formulation optimizes maximizing the cosine similarity between positive pairs and minimizing the cosine similarity between negative pairs. The loss is weighted by the weighting mechanism as shown in the algorithm.

---
**Algorithm 1:** Training Loop for the ReweightOOD framework
---
**Data:** Training data: $X, Y$ (labels)
**Result:** Trained model parameters: $f_\theta$
1 Initialize model parameters $f_\theta$ and projection head $g_\theta$ ;
2 Define learning rate $\alpha$;
3 Define linear transformation constant $m_w, c_w, m_b, c_b$;
4 Define the number of epochs $N$;
5 **for** $epoch \leftarrow 1$ **to** $N$ **do**
6    **for** *each* $(x_i, y_i)$ *in* $X, Y$ **do**
7      Apply random augmentations $x_i = RandomAugment(x_i)$;
8      Compute encoded representation $h_i = f_\theta(g_\theta(x_i))$;
9      Transform to hyperspherical embeddings $\hat{h}_i = L2\_Normalize(h_i)$;
10      similarity_matrix = embeddings @ embeddings.transpose$(1, 0)$ ;
11      label_matrix = label.unsqueeze(1) == label.unsqueeze(0) ;
12      within_class_matrix = label_matrix.triu(diagonal=1) ;
13      between_class_matrix = label_matrix.logical_not().triu(diagonal=1);
14      similarity_matrix = similarity_matrix.view(-1) ;
15      within_class_matrix = within_class_matrix.view(-1);
16      between_class_matrix = between_class_matrix.view(-1);
17      $s_w^j$ = similarity_matrix[within_class_matrix] ;
18      $s_b^i$ = similarity_matrix[between_class_matrix];
19      $w_w^j = sigmoid(m_w \cdot s_w + c_w)$ ;
20      $w_b^i = sigmoid(m_b \cdot s_b + c_b)$ ;
21      loss $= \log\left(\sum_{i=1}^n \exp(w_b^i \cdot s_b^i/\tau)\right) - \log\left(\sum_{j=1}^m \exp(w_w^j \cdot s_w^j/\tau)\right)$ ;
22      Compute gradients with respect to model parameters ;
23      Update learnable parameters: $\theta \leftarrow \theta - \alpha\nabla L_i$;
24 **return** Trained model parameters $f_\theta$;
---

## .2 ABLATION ON LINEAR TRANSFORMATION

To investigate the effect of linear transformation prior to sigmoid weighting, we compare the use of linear transformation with the use of the original cosine similarity range for both within-class and between-class similarity. The results in Table 1 (FPR metric) and Table 2 (AUROC metric) show that the use of linear transformation as opposed to just using original cosine similarity enhances the performance of OOD detection.

Table 1: Ablation on use of linear transformation in the weighting of ($s_w$, $s_b$) and OOD detection performance in terms of FPR metric.

| Method | OOD Dataset | | | | | | | |
|---|---|---|---|---|---|---|---|---|
| | MNIST | iSUN | LSUN | LSUN-r | SVHN | Texture | Places365 | **Average** |
| Without linear transformation (Original range) | 64.47 | 77.14 | 34.46 | 75.42 | 17.54 | 33.40 | 80.72 | 54.74 |
| Without linear transformation in $s_w$ weighting | 68.79 | 56.80 | 22.18 | 56.49 | 12.61 | 37.82 | 78.79 | 47.64 |
| Without linear transformation in $s_b$ weighting | 48.67 | 74.53 | 39.23 | 70.48 | 14.83 | 31.86 | 77.65 | 51.04 |
| **(ReweightOOD) *Ours*** | 19.24 | 57.56 | 19.59 | 56.31 | 8.39 | 28.72 | 78.70 | **38.36** |

Table 2: Ablation on use of linear transformation in weighting of $(s_w, s_b)$ and OOD detection performance in terms of AUROC metric.

| Method | OOD Dataset | | | | | | | |
|---|---|---|---|---|---|---|---|---|
| | MNIST | iSUN | LSUN | LSUN-r | SVHN | Texture | Places365 | Average |
| Without linear transformation (Original range) | 84.16 | 81.75 | 90.47 | 83.01 | 96.16 | 92.47 | 75.85 | 86.27 |
| Without linear transformation in $s_w$ weighting | 87.98 | 88.35 | 95.67 | 88.92 | 97.72 | 92.62 | 76.52 | 89.68 |
| Without linear transformation in $s_b$ weighting | 89.00 | 82.85 | 90.47 | 84.26 | 96.95 | 92.85 | 77.94 | 87.76 |
| **(ReweightOOD)** *Ours* | 96.86 | 87.54 | 96.86 | 88.23 | 98.31 | 94.11 | 76.01 | **90.91** |

## .3 STABILITY OF REWEIGHTOOD

We perform 3 independent trials using ResNet18 network to verify strong performance of ReweightOOD in comparison to baseline with CIFAR100 datasets. The OOD detection performance metrics in FPR metrics in the form of mean$_{\pm\text{std}}$ presented in Table 3 verifies effectiveness of our approach.

Table 3: OOD detection performance across 3 independent trials using ResNet18 trained with CIFAR100 datasets in FPR metric.

| ID dataset | OOD Dataset | | | | | | | |
|---|---|---|---|---|---|---|---|---|
| | MNIST | iSUN | LSUN | LSUN-r | SVHN | Texture | Places365 | Average |
| Baseline | $72.58_{\pm7.01}$ | $74.95_{\pm14.62}$ | $46.05_{\pm7.58}$ | $72.00_{\pm13.79}$ | $25.08_{\pm1.70}$ | $42.33_{\pm3.08}$ | $71.56_{\pm2.18}$ | $57.79_{\pm5.53}$ |
| **(ReweightOOD)** *Ours* | $16.84_{\pm5.99}$ | $68.05_{\pm8.10}$ | $24.06_{\pm9.71}$ | $67.21_{\pm8.11}$ | $9.01_{\pm2.03}$ | $27.36_{\pm2.76}$ | $79.59_{\pm2.08}$ | $\mathbf{41.73_{\pm2.56}}$ |

## .4 ADDITIONAL CIFAR100 RESULTS

The OOD detection performance in various OOD datasets obtained with DenseNet architecture trained with the CIFAR-100 dataset in terms of FPR metric is summarized in Table 4. The superior performance of ReweightOOD is evident from the presented results.

Table 4: OOD detection performance using CIFAR-100(ID) with DenseNet architecture in terms of FPR.

| Method | OOD Dataset | | | | | | | |
|---|---|---|---|---|---|---|---|---|
| | MNIST | iSUN | LSUN | LSUN-r | SVHN | Texture | Places365 | Average FPR ↓ |
| Baseline | 37.32 | 39.66 | 35.63 | 46.58 | 11.15 | 20.43 | 82.45 | 39.03 |
| SupCon | 26.97 | 61.00 | 30.60 | 70.02 | 9.93 | 22.13 | 89.34 | 44.28 |
| **(ReweightOOD)** *Ours* | 10.39 | 24.45 | 36.86 | 29.96 | 14.26 | 18.53 | 85.08 | **31.36** |

## .5 CIFAR10 RESULTS

For CIFAR10 (ID) experiments, the comparison of our method with various non-contrastive as well as contrastive approaches is presented in Table 5. In the non-contrastive approach, We compare our approach with posthoc methods such as MSP (Hendrycks & Gimpel, 2017), ODIN (Liang et al., 2017), Mahalanobis (Lee et al., 2017), DICE Sun & Li (2022) and Energy (Liu et al., 2020) as well as training-time regularization approaches such as GODIN (Hsu et al., 2020) and LogitNorm (Wei et al., 2022). In contrastive approaches, we compare with ProxyAnchor (Kim et al., 2020), CSI (Tack et al., 2020), SSD+ (Sehwag et al., 2021), KNN+ (Sun et al., 2022), and CIDER (Ming et al., 2023). In contrastive approaches, we compare our method with ProxyAnchor (Kim et al., 2020), CSI (Tack et al., 2020), SSD+ (Sehwag et al., 2021), KNN+ (Sun et al., 2022), and CIDER (Ming et al., 2023).We use default hyperparameters provided in the original work whenever required. Our approach leads to the best performance in both metrics. We can observe that the majority of the methods work well since it is a relatively easier task because of the smaller number of classes in comparison to CIFAR100. However, our approach still outperforms the current approaches on average FPR and AUROC metrics.

Table 5: Mean OOD detection performance for CIFAR-10 (ID) with ResNet-18.

| Method | MNIST | | iSUN | | LSUN | | LSUN-r | | SVHN | | Texture | | Places365 | | Average | |
|---|---|---|---|---|---|---|---|---|---|---|---|---|---|---|---|---|
| | FPR↓ | AUROC↑ | FPR↓ | AUROC↑ | FPR↓ | AUROC↑ | FPR↓ | AUROC↑ | FPR↓ | AUROC↑ | FPR↓ | AUROC↑ | FPR↓ | AUROC↑ | FPR↓ | AUROC↑ |
| **Without Contrastive Learning** | | | | | | | | | | | | | | | | |
| MSP | 45.93 | 93.02 | 47.42 | 92.07 | 30.90 | 95.49 | 45.77 | 92.78 | 46.76 | 92.33 | 52.82 | 90.44 | 55.05 | 89.03 | 46.38 | 92.17 |
| ODIN | 20.45 | 95.37 | 24.96 | 93.32 | 16.33 | 95.93 | 22.64 | 94.31 | 51.00 | 84.36 | 41.93 | 87.35 | 50.17 | 84.35 | 32.50 | 90.71 |
| Mahalanobis | 77.84 | 88.43 | 30.44 | 94.67 | 79.41 | 86.19 | 30.54 | 94.80 | 64.60 | 90.21 | 36.05 | 93.29 | 58.53 | 88.42 | 53.92 | 90.86 |
| Energy | 23.77 | 95.20 | 36.45 | 91.65 | 13.04 | 97.36 | 33.91 | 92.78 | 33.16 | 92.49 | 41.77 | 90.26 | 42.63 | 89.15 | 32.11 | 92.70 |
| DICE | 12.57 | 97.63 | 73.46 | 80.54 | 12.37 | 97.29 | 72.14 | 81.75 | 30.48 | 93.93 | 58.81 | 82.55 | 64.32 | 78.87 | 46.31 | 87.51 |
| GODIN | 5.53 | 98.77 | 12.48 | 97.55 | 6.55 | 98.64 | 10.73 | 97.88 | 14.39 | 97.62 | 20.59 | 96.05 | 40.89 | 90.24 | 15.88 | 96.68 |
| LogitNorm | 3.16 | 99.16 | 23.87 | 96.23 | 0.47 | 99.83 | 20.84 | 96.72 | 13.71 | 97.63 | 27.94 | 95.10 | 30.29 | 94.92 | 17.18 | 97.08 |
| **With Contrastive Learning** | | | | | | | | | | | | | | | | |
| ProxyAnchor | 7.77 | 98.72 | 26.13 | 96.09 | 4.26 | 98.99 | 21.36 | 96.54 | 1.04 | 99.80 | 9.68 | 98.44 | 19.20 | 96.51 | 12.78 | 97.87 |
| CSI | 4.20 | 98.32 | 45.36 | 92.33 | 20.62 | 97.07 | 47.07 | 92.16 | 4.58 | 99.02 | 14.66 | 97.66 | 58.67 | 88.71 | 27.88 | 95.04 |
| SSD+ | 2.74 | 99.34 | 54.62 | 93.16 | 5.04 | 99.05 | 52.17 | 93.55 | 0.45 | 99.90 | 9.08 | 98.49 | 24.15 | 96.22 | 21.18 | 97.10 |
| KNN+ | 4.95 | 99.09 | 21.23 | 96.53 | 4.52 | 99.16 | 21.83 | 96.44 | 1.74 | 99.64 | 8.88 | 98.48 | 21.28 | 96.43 | 12.06 | 97.97 |
| CIDER | 31.03 | 94.97 | 13.67 | 97.36 | 4.07 | 99.26 | 14.40 | 97.31 | 2.85 | 99.54 | 18.55 | 97.15 | 30.27 | 95.17 | 16.41 | 97.25 |
| Baseline | 10.25 | 98.37 | 18.12 | 96.99 | 8.51 | 98.59 | 17.20 | 97.14 | 1.43 | 99.75 | 9.11 | 98.51 | 24.04 | 96.07 | 12.67 | 97.92 |
| **(ReweightOOD)** *Ours* | 10.99 | 97.92 | 10.85 | 98.00 | 3.42 | 99.23 | 10.99 | 97.94 | 1.03 | 99.82 | 9.75 | 98.47 | 22.72 | 95.97 | **9.96** | **98.19** |

### .6 COMPATIBILITY WITH MAHALONOBIS DISTANCE CIFAR EXPERIMENTS.

The OOD detection score using Mahalanobis distance using CIFAR-10 experiments is shown in Table 6 and Table 7. Also, the OOD detection performance using CIFAR-100 experiments in terms of the AUROC metric is also shown in Table 8. Experiments are performed using the ResNet-18 network. The results presented show that our approach consistently remains superior in OOD detection performance.

Table 6: Compatibility with MDS using CIFAR-10 (ID) in terms of FPR metric.

| Method | iSUN | LSUN | LSUN-r | SVHN | Texture | Places365 | Average FPR |
|---|---|---|---|---|---|---|---|
| ProxyAnchor | 55.83 | 4.42 | 51.43 | 0.14 | 13.37 | 24.60 | 24.96 |
| SSD+ | 46.60 | 2.56 | 42.16 | 0.28 | 8.67 | 22.23 | 20.42 |
| CIDER | 53.32 | 17.17 | 51.27 | 0.86 | 22.30 | 45.06 | 31.66 |
| **(ReweightOOD)** *Ours* | 30.67 | 6.26 | 26.91 | 0.10 | 8.59 | 22.64 | **15.86** |

Table 7: Compatibility with MDS using CIFAR-10 (ID) in terms of AUROC metric.

| Method | iSUN | LSUN | LSUN-r | SVHN | Texture | Places365 | Average FPR |
|---|---|---|---|---|---|---|---|
| ProxyAnchor | 92.48 | 99.10 | 93.19 | 99.95 | 97.92 | 96.08 | 96.45 |
| SSD+ | 93.90 | 99.36 | 94.46 | 99.93 | 98.56 | 96.36 | 97.10 |
| CIDER | 93.74 | 97.77 | 93.86 | 99.80 | 96.56 | 93.70 | 95.91 |
| **(ReweightOOD)** *Ours* | 95.65 | 98.80 | 96.07 | 99.96 | 98.50 | 96.31 | **97.55** |

Table 8: Compatibility with MDS using CIFAR-100 (ID) in terms of AUROC metric.

| Method | MNIST | iSUN | LSUN | LSUN-r | SVHN | Texture | Places365 | Average AUROC ↑ |
|---|---|---|---|---|---|---|---|---|
| ProxyAnchor | 76.32 | 75.41 | 90.23 | 75.81 | 98.47 | 87.08 | 82.66 | 83.71 |
| SSD+ | 76.80 | 83.85 | 89.91 | 85.19 | 95.70 | 88.22 | 80.86 | 85.79 |
| CIDER | 75.78 | 79.17 | 95.98 | 79.38 | 97.99 | 90.10 | 80.30 | 85.53 |
| **(ReweightOOD)** *Ours* | 87.59 | 86.30 | 96.68 | 86.79 | 98.81 | 91.01 | 81.70 | **89.84** |

### .7 OOD DETECTION PERFORMANCE IN IMAGENET DATASET IN TERMS OF AUROC METRICS.

Further OOD detection performance in ImageNet100 experiments in terms of AUROC is presented in Table 9. Finetuning pretrained ResNet50 model with our approach leads to the superior AUROC metric in comparison to baseline and SupCon loss formulations. This shows the potential of ReweightOOD to be used in the fine-tuning context too.

### .8 IMPLEMENTATION PLATFORM

All the experiments were performed on NVIDIA A100 GPU with PyTorch deep learning library (version: 1.8).

Table 9: OOD detection performance in large-scale experiments (ImageNet-100) in terms of AU-ROC by fine-tuning pretrained ResNet50.

| Method | iNaturalist | SUN | Textures | Places | SSB Hard | Ninco | Openimage | Average **AUROC** ↓ |
|---|---|---|---|---|---|---|---|---|
| Baseline | 99.15 | 99.35 | 99.12 | 98.56 | 92.17 | 94.46 | 98.47 | 97.32 |
| SupCon | 99.15 | 99.35 | 99.12 | 98.56 | 92.17 | 95.46 | 98.47 | 97.32 |
| **(ReweightOOD)** *Ours* | 99.36 | 99.26 | 99.45 | 98.49 | 92.29 | 95.16 | 98.81 | **97.55** |

## .9 DATASETS

### .9.1 ID DATASETS

CIFAR-10, CIFAR-100 and ImageNet-100 are three ID datasets used in our experiments.

**CIFAR** CIFAR-10 is one of the most commonly used datasets for benchmarking computer vision performance, especially for classification tasks. It contains 10 categories of images. There are 50,000 training and 10,000 testing samples totalling 60,000 samples. CIFAR-100 is a very similar dataset to CIFAR-10 but consists of 100 classes. It contains the same number of images as CIFAR-10, however number of images per category is smaller. The CIFAR images exhibit a square resolution with a width of 32 pixels.

**ImageNet100** ImageNet-100 dataset is a subset of ImageNet dataset. ImageNet-100 consists of images from 100 randomly sampled categories from the original ImageNet dataset. The images of the ImageNet100 dataset are trained in the square resolution with a width of 224 pixels.

### .9.2 OOD DATASETS

We use the following OOD datasets: MNIST, iSUN, LSUN-R, LSUN-C, Places365, SVHN, Texture, iNaturalists, NINCO, OpenImage, and Semantic Shift Benchmark (SSB).

**MNIST** MNIST dataset consists of 70,000 grayscale images. Each image represents a handwritten digit ranging from 0 to 9 in a resolution of 28x28 pixels. MNIST dataset consists of 60,000 training 10,000 testing images.

**SVHN** SVHN is a real-world digit recognition dataset obtained from house numbers in Google Street View images. It is similar to MNIST images but the difficulty of recognition for machine learning algorithms is a more harder.

**LSUN** LSUN datasets are curated for the purpose of scene understanding. It has three variations.

**Places365** Places365 is also another large-scale scene dataset developed for training deep-learning models to understand scenes.

**Texture** The Textures dataset contains images of various textures which are unique images apart from widely available object or scene images.

**iNaturalist** iNaturalist contains images of the natural world. It has 13 super-categories and 5,089 sub-categories covering various aspects of natural realms such as plants, insects, birds, mammals, etc.

**NINCO** The NINCO (No ImageNet Class Objects) dataset comprises 64 OOD classes, encompassing a total of 5879 samples. These OOD classes were thoughtfully chosen to ensure the complete absence of categorical overlap with any of the 1000 classes found in ImageNet-1K. Subsequently, a rigorous individual inspection of each sample was conducted to verify the absence of any objects from the ID categories.

**SSB** SSB is a dataset for open-set recognition, category discovery, out-of-distribution detection, etc. The SSB is intended to isolate semantic novelty from other forms of distributional shifts. It contains easy and hard split, but we make use of hard split.

**OpenImage-O** OpenImage-O is curated by selecting images on an individual basis from the test set of OpenImage-V3, which, in turn, was taken from the vast repository of Flickr without the imposition of a predetermined list of class names or tags. This dataset is manually annotated and has a diverse distribution.

In all OOD datasets, samples duplicating in ID datasets are removed wherever needed hence obtaining subset the subset different from ID.