# OpenReview forum: "ReweightOOD: Loss Reweighting for Distance-based OOD Detection"
_ICLR.cc/2024/Conference — Submitted to ICLR 2024_

### Official Review · Reviewer_8LLh · 2023-10-16

**Soundness:** 2 fair
**Presentation:** 3 good
**Contribution:** 1 poor
**Rating:** 3
**Confidence:** 4

**Summary:**

Out-of-distribution (OOD) detection has been widely studied. This paper focuses on the distance-based OOD detection with contrastive optimization methods. It points out that assigning equal significance to all similar pairs is not efficient in reducing the MES for each class and achieving higher inter-class dispersion. Therefore, this paper proposes ReweightOOD, a weight mapping function based on similarity, to prioritize hard positives and hard negatives. Experiments and visualization results show the proposed ReweightOOD surpasses the baseline and SOTAs by a large margin on FPR and AUROC.

**Strengths:**

Based on the distance-based methods in OOD detection, this paper may first combine reweighting the hard samples with contrastive optimization.

The paper conducts explicit experiments on the influence the ReweightOOD brings to MES and inter-class dispersion. The results on CIFAR and ImageNet also verify its effectiveness.

**Weaknesses:**

The novelty of this idea is quite limited. Designing different weights for hard samples has been widely applied in many fields including classification, detection, etc., and yields success. It can be directly accustomed to nearly all methods and will not result in depreciation at least.

The baseline chosen in this paper training a not-well-adjusted model to perform OOD Detection is quite unfair. As the proposed method is a loss function, it can be combined with other contrastive optimization methods easily. A method designed for OOD detection is more suitable to be a baseline. Therefore, it makes the results less convincing.

The important hyperparameters including the scaling and shifting scalars of the final linear transformation are not studied in the ablation experiments.

**Questions:**

Why are the parameters of two linear transformation layers directly set to (5, -4, 2, 1) or (5, -2, 2, 1)?

What is the reason for using a smaller shifting scalar from -2 to -4 when training with a smaller dataset CIFAR10?

---

> ### Author Response · Authors · 2023-11-20
> **Response to reviewer 8LLh**
>
> We thank reviewer for their presented concerns which we address them below.
>
> It's important to note that the primary focus is to demonstrate the effectiveness of our proposed method in OOD detection. Furthermore, we study the reweighting approach’s efficacy for the first time in OOD detection. Such work in the domain has been missing, which makes our work novel in OOD detection.
>
> We perform the comparison of our method with various approaches well suited for OOD detection. The key takeaway lies in the superior performance of our approach compared to competitive approaches like SupCon, CSI, and CIDER. Moreover, as presented in the paper, our work demonstrates the distinctively remarkable performance in comparison to strong post-hoc methods too. As requested by reviewer XxBj, we haver updated the manuscript with additional posthoc methods.
>
> The hyperparameters are determined from the validation set.
>
> The sensitivity study of these hyperparameters using ResNet18 network with CIFAR100 datasets is presented below.
>
>
> |$m_b$ | Average FPR | &#124; | $c_b$ | Average FPR | &#124; | $m_w$ | Average FPR | &#124; | $c_w$ | Average FPR | &#124;
> |----|--------------|-------|-----|--------------|-------|-----|--------------|-------|-----|--------------|-------|
> | 4  | 51.01        | &#124;| 1   | 44.96        | &#124;| 1   | 69.10        | &#124;| 1   | 38.36        | &#124;|
> | 5  | 38.36        | &#124;| 2   | 38.36        | &#124;| 2   | 38.36        | &#124;| 2   | 67.63        | &#124;|
> | 6  | 39.14        | &#124;| 3   | 52.66        | &#124;| 3   | 43.44        | &#124;| 3   | 70.84        | &#124;|

---

### Official Review · Reviewer_tSwH · 2023-10-30

**Soundness:** 2 fair
**Presentation:** 3 good
**Contribution:** 2 fair
**Rating:** 3
**Confidence:** 4

**Summary:**

This paper uses contrastive learning to address out-of-distribution (OOD) problems. The article introduces what contrastive learning is and discusses the challenges encountered by classical contrastive learning in tackling OOD problems. The authors believe that the main issue with contrastive learning in addressing OOD problems is that the discrimination between many difficult samples is not high enough, resulting in a close proximity among these challenging samples. As a result, it becomes difficult to recognize OOD samples. Therefore, the authors propose a reweighting approach to enhance the discrimination between different categories in contrastive learning, thereby improving the effectiveness of OOD detection.

**Strengths:**

1. The method proposed in this paper is simple and intuitive, and may be a good way to solve OOD problems.
2. According to the author's experimental results, the proposed method can indeed improve the performance of OOD detection.
3. The author's writing is very clear, and the description of the method section is easy to understand, combining formulas and diagrams.

**Weaknesses:**

1. In the abstract, the author emphasizes that their method outperforms the baseline by 38%, which seems impressive. However, this improvement is mainly due to the low performance of the baseline method itself. The proposed method does not actually achieve such a significant improvement compared to the state-of-the-art (SOTA). In my opinion, this exaggerates the contribution of this paper. It would be better to clarify the improvement relative to SOTA in the beginning to avoid misleading the readers.

2. The simplicity of the method itself can be considered an advantage. However, based on the experimental results, the improvement of this paper compared to the state-of-the-art is limited. Therefore, I hope the authors can further explore this direction and achieve more substantial advancements.

3. The experiments conducted in this paper are not comprehensive enough. Firstly, experiments on complete ImageNet-1K were not conducted. Additionally, the comparison with the latest methods, such as those presented in recent conferences like CVPR 2023, was not included.

**Questions:**

Please address the problem in weakness.

---

> ### Author Response · Authors · 2023-11-20
> **Response to Reviewer tSwH.**
>
> We thank the reviewer for the suggestions. We address the concerns below.
>
> - We will clarify the improvement relative to the SoTA approaches by being more specific in the abstract in the upcoming versions. We believe the performance improvement compared to the current competitive approaches such as SupCon and CIDER is still significant.
>
> - Orthogonal to the papers presented in CVPR2023, our work has the distinction in that it adopts a reweighting approach which is novel to the best of our knowledge.
>
> - The ImageNet-1k experiments adopting the ImageNet100 setup is presented below:
>
>   | Method                 | iNaturalist | SUN  | Textures | Places | SSB Hard | Ninco | Openimage | Average **FPR** $\downarrow$ |
>   |------------------------|-------------|------|----------|--------|----------|-------|-----------|--------------------------------|
>   | SupCon                 | 61.93        | 68.85 | 25.43     | 75.28   | 90.79    | 78.82 | 55.64      | 65.25                          |
>   | CIDER                   | 53.54        | 61.27 | 23.95     | 68.76   | 91.52    | 79.15 | 54.22      | 61.77                          |
>   | **(ReweightOOD) *Ours***| 50.70        | 59.46 | 24.27     | 66.27   | 90.18   | 76.76 | 50.97      | **59.80**                      |
>
>   It can be observed that the proposed approach has noticeable gains over SupCon and CIDER demonstrating its superiority.

---

### Official Review · Reviewer_XxBj · 2023-11-01

**Soundness:** 2 fair
**Presentation:** 2 fair
**Contribution:** 1 poor
**Rating:** 3
**Confidence:** 5

**Summary:**

This paper proposes some modifications to contrastive optimization-based OOD detection. Previous approaches assign equal importance to all similar instances. This paper proposes to give different weights to these instances and tries to enforce the minimum enclosing sphere and higher inter-class dispersions. Experiments on CIFAR100 and ImageNet 100 demonstrate the effectiveness of the proposed approach.

**Strengths:**

1. The proposed method is very simple and intuitively makes sense. The idea of obtaining a minimum closure sphere makes sense to separate OOD and ID data, and it is also easy to implement this idea by adding weights to the loss functions.

**Weaknesses:**

1. **There are many recent OOD baselines but they are not taken into consideration for comparison.** There are many recent OOD baselines but the authors either do not cite them [1,2,3] or do not add them in the comparison (e.g., ASH, GradNorm). In particular, these are very strong baselines and I see the performance of this method at the same level as ReAct. It is a bit strange that the evaluation does not involve any activation-clipping baselines such as ReAct and ASH (they are clear state-of-the-art methods and do not require any training).


>[1] React: Out-of-distribution detection with rectified activations. NeurIPS 2021.
>
>[2] RankFeat: Rank-1 Feature Removal for Out-of-distribution Detection. NeurIPS 2022.
>
>[3] Boosting Out-of-distribution Detection with Typical Features. NeurIPS 2022.


2. **The experiments are not sufficient only on CIFAR/ImageNet100.** The authors only validate the proposed approach on CIFAR100 and ImageNet100, which is far from sufficient and comprehensive. Usually, people evaluate OOD methods on CIFAR10, CIFAR100, and ImageNet-1k benchmarks. I would be convinced if the method also works for ImageNet-1k.

3. **The instance weight can be replaced with learnable temperatures.**  Adding weight significance to the samples is one way to obtain a higher inter-class separation, but I am wondering if would it also make sense to make the temperature learnable. Compared with weight significance, it would bring another benefit: learnable distribution shaping for each category.

4. **How does the method perform when scaling to more/fewer categories?** I would suspect the performance is highly related to the number of categories. If the value is too large (e.g., 1000) or too small (e.g., 5), the performance might deteriorate. Can authors provide some ablation studies on subsets of the used datasets?

5. **Does the method also work for Transformer-based architectures?** Currently the authors evaluate their approach with ResNet-18, DenseNet and WideResNet? Can authors do more experimental evaluation on Transformers?

**Questions:**

Please see the weaknesses.

---

> ### Author Response · Authors · 2023-11-20
> **Response to reviewer XxBj.**
>
> We thank the reviewer for the concrete suggestions. We address the presented concerns below:
>
> - The updated manuscript now contains the suggested comparisons and we do cite all the mentioned works in the manuscript.
>
> - The experiments with the Imagenet-1k dataset following the setup of the ImageNet-100 experiment are shown below.
>   | Method                 | iNaturalist | SUN  | Textures | Places | SSB Hard | Ninco | Openimage | Average **FPR** $\downarrow$ |
>   |------------------------|-------------|------|----------|--------|----------|-------|-----------|--------------------------------|
>   | SupCon                 | 61.93        | 68.85 | 25.43     | 75.28   | 90.79    | 78.82 | 55.64      | 65.25                          |
>   | CIDER                   | 53.54        | 61.27 | 23.95     | 68.76   | 91.52    | 79.15 | 54.22      | 61.77                          |
>   | **(ReweightOOD) *Ours***| 50.70        | 59.46 | 24.27     | 66.27   | 90.18   | 76.76 | 50.97      | **59.80**                      |
>
>   It can be observed that the proposed approach has noticeable gains over SupCon and CIDER demonstrating its superiority.
>
> - We thank the reviewer for the possible further extensions. We aim to undertake such an approach in future work.
>
> - As can be observed from the above ImageNet-1k experiments, our approach demonstrates performance gain even when scaled to an even more challenging (higher number of categories) setting.

---

### Official Review · Reviewer_NpEj · 2023-11-02

**Soundness:** 3 good
**Presentation:** 2 fair
**Contribution:** 2 fair
**Rating:** 5
**Confidence:** 4

**Summary:**

The traditional distance-based OOD detection assumes that OOD samples are far from In-Distribution (ID) clusters in the embedding space. A recent method involves contrastive optimization to create an OOD-detection-friendly embedding space. However, this approach doesn't effectively reduce the Minimum Enclosing Sphere (MES) for each class and achieve higher inter-class dispersion, leading to potential overlap between ID and OOD samples. To address this, the paper proposes a reweighting scheme called ReweightOOD. This scheme prioritizes optimizing less-optimized contrasting pairs while assigning lower importance to already well-optimized pairs.

**Strengths:**

This paper addresses the important issue of Out-of-Distribution (OOD) detection in neural networks, crucial for ensuring safety and reliability in critical applications.

**Weaknesses:**

1. The figures 3 and 4 are not mentioned in the article.
2. I find Figure 5 visually unappealing, and I wonder why there is only one category in the right figure.
3. Table 2 only has two columns, it's not worth occupying such a large space in the paper.
4. More importantly, why not conduct experiments on the ImageNet dataset, like existing works [1] have done?

[1] Sun Y, Guo C, Li Y. React: Out-of-distribution detection with rectified activations[J]. Advances in Neural Information Processing Systems, 2021, 34: 144-157.

**Questions:**

See above.

---

> ### Author Response · Authors · 2023-11-20
> **Response to Reviewer NpEj.**
>
> We thank Review NpEj for the presented concerns. We address the concerns below.
>
> - We have resolved the issue in the latest manuscript by referencing Figure 3 and Figure 4.
>
> - Apologies for the overleaf rendering issues, the correctly rendered visualization is updated in the manuscript.
>
> - Thanks for noticing the space consideration. We'll be mindful of it in the subsequent version.
>
> - Thanks for pointing it out. Adopting the training setup of ImageNet100, we present the requested comparison with ImageNet-1k below:
>
>   | Method                 | iNaturalist | SUN  | Textures | Places | SSB Hard | Ninco | Openimage | Average **FPR** $\downarrow$ |
>   |------------------------|-------------|------|----------|--------|----------|-------|-----------|--------------------------------|
>   | SupCon                 | 61.93        | 68.85 | 25.43     | 75.28   | 90.79    | 78.82 | 55.64      | 65.25                          |
>   | CIDER                   | 53.54        | 61.27 | 23.95     | 68.76   | 91.52    | 79.15 | 54.22      | 61.77                          |
>   | **(ReweightOOD) *Ours***| 50.70        | 59.46 | 24.27     | 66.27   | 90.18   | 76.76 | 50.97      | **59.80**                      |
>
>   It can be observed that the proposed approach has noticeable gains over SupCon and CIDER demonstrating its superiority.

---

### Official Review · Reviewer_4EKT · 2023-11-10

**Soundness:** 3 good
**Presentation:** 3 good
**Contribution:** 3 good
**Rating:** 6
**Confidence:** 4

**Summary:**

This paper tackles the problem of OOD detection. It is motivated from supervised contrastive learning and propose to further disperse the inter-class distance while reduce intra-class variance via reweighting. Specifically, it encourages the model to focus more on hard positives and negatives samples. The experimenets on several benchmarks show the performance advantages of the method.

**Strengths:**

- The motivation is clear and presentation is easy to follow.
- The proposed method is straightforward and weights for positives and negatives are dynatmically adjusted based on the similarity score.
- Table 1 and 2 shows the dispersion cross class and compactness within the class for the proposed method compared to SupCon
- Performance gains on CIFAR10/100 and ImageNet100

**Weaknesses:**

Novelty:
- Although the reweighting on constrative learning has been under-explored in OOD detection community, focusing more on hard postivie/negatives are widely applied in deep metric learning (especially in face recognition). E,g [1][2]. I think the novelty is sort of limited as the paper adopt hard sample mining/weighting into OOD detection task. In rebuttal, can you explain the difference bewteen your work and [1,2] in terms of hard samples re-weighting?

Experiment:
- In ImageNet100, can we have numbers from other methods (e.g, CIDER) instead of SupCon only.
- For Table 1,2, I am interested to see how the MES score and Average centroid dispersion for CIDER.
- Hyperparameters test on linear transformation are encouraged. It is said that ResNet18 network are set to (5, −2, 2, 1) and (5, −4, 2, 1). Do you have to set differently for other backbones or datasets?
- For Figure 5 (b), why there is only one cluster? also, can you provide feature vislization for SupCon or CIDER as well?

[1] CurricularFace: Adaptive Curriculum Learning Loss for Deep Face Recognition
[2] AdaFace: Quality Adaptive Margin for Face Recognition

**Questions:**

Please refer to the weakness

---

> ### Author Response · Authors · 2023-11-12
> **Response to Reviewer 4EKT.**
>
> We thank Reviewer 4EKT for raising insightful concerns and queries. We address them below.
>
> A detailed clarification of the distinction between our proposed approach against CurricularFace and AdaFace is presented below.
>
> -  CurricularFace adopts a curriculum-style learning approach, wherein early/late training dynamics are taken into account. It emphasizes easy samples at the earlier stage while prioritizing the hard samples at the later stage. ReweightOOD does not impose curriculum-style learning at all, as it does not distinguish between early and late training phases. The reweighting function of ReweightOOD, in the form of a sigmoid, strategically suppresses easy samples from the outset, directing attention toward challenging samples. AdaFace, another proxy-based approach, adopts the quality of the image -- approximated by feature norm -- into the weighting of the sample's importance. In AdaFace, hard samples from higher-quality images receive increased weighting, while easy samples from lower-quality images are weighted higher. In contrast, our approach focuses solely on the cosine similarity when reweighting pair instances. Moreover, ReweightOOD facilitates distinct treatment of positive and negative pairs due to the presence of a transformation function with separate hyperparameters. The utilization of two sets of linear transformations, specific to positive and negative pairs, enables flexible weighting tailored to achieving optimal OOD detection. Additionally, both curricularFace and adaFace adopt cross-entropy-based optimization while ReweightOOD uses similarity maximization/minimization optimization as shown in equation (3).
>
> Regarding the experiments, we address the queries below:
>
> - The result of ImageNet100 including CIDER is presented below:-
>
>   | Method                 | iNaturalist | SUN  | Textures | Places | SSB Hard | Ninco | Openimage | Average **FPR** $\downarrow$ |
>   |------------------------|-------------|------|----------|--------|----------|-------|-----------|--------------------------------|
>   | Baseline               | 3.07        | 2.39 | 4.57     | 5.47   | 35.39    | 29.15 | 7.05      | 12.44                          |
>   | SupCon                 | 2.43        | 1.98 | 2.59     | 5.43   | 34.25    | 25.58 | 5.28      | 11.08                          |
>   | CIDER                  | 5.07        | 1.85 | 1.77     | 5.70   | 37.35    | 27.93 | 5.73      | 12.20                          |
>   | **(ReweightOOD) *Ours***| 2.18        | 1.97 | 2.73     | 5.29   | 32.00    | 24.63 | 5.06      | **10.55**                      |
>
>   As can be observed from the above Table, our approach yields the minimum average FPR of 10.55.
>
>
> - The updated MES Table with the results of CIDER is given below:
>    | Method                 | Apples | Aquarium Fish | Baby | Bear | Beaver | Bed  | Bee | Beetle | Bicycle | Bottles | ... | Mean  |
>    |------------------------|--------|---------------|------|------|--------|------|-----|--------|---------|---------|-----|-------|
>    | Baseline               | 1.09   | 1.11          | 1.07 | 0.93 | 0.97   | 1.07 | 1.03| 1.07   | 1.10    | 1.07    | ... | 1.05  |
>    | SupCon                 | 0.97   | 1.01          | 0.97 | 1.00 | 1.00   | 0.98 | 0.99| 0.97   | 1.04    | 1.03    | ... | 1.01  |
>    |  *CIDER*              | 0.96   | 0.94          | 0.89 | 0.91 | 0.86   | 0.88 | 0.87| 0.88   | 1.04    | 0.99    | ... | 0.92  |
>    | **(ReweightOOD) *Ours***| 0.95   | 0.90          | 0.90 | 0.89 | 0.84   | 0.90 | 0.89| 0.91   | 0.99    | 0.96    | ... | **0.90** |
>
>    As evidenced from the above Table, the average MES of ReweightOOD is minimum among all. Moreover, the class centroid dispersion of ReweightOOD (0.63) is comparable to that of CIDER (0.64) despite not explicitly encouraging the inter-class dispersion.
>
> - The random hyperparameter optimization is employed with respect to the validation set (gaussian noise). The hyperparameter set to (5, −4, 2, 1) consistently demonstrates strong performance in experiments conducted on both CIFAR100 and ImageNet100 datasets, across all architectures used in the paper.
>
> - Apologies for the overleaf rendering issues, the correctly rendered visualization is updated in the manuscript.

---

### Meta-Review · Area_Chair_PnQ5 · 2023-11-26

**Metareview:**

This paper tackles the problem of OOD detection. Motivated by supervised contrastive learning, a reweighting-based approach proposed to enforce the model focus more on hard positives and negatives. Experiments show the advantages of the proposed method over previous methods.

**Strengths**
- The motivation of this paper is clear. The proposed method is simple and reasonable.
- Despite simple, the proposed method can improve the performance of OOD detection.

**Weaknesses**

- Lack of comparison with many recent works.
- The novelty of this idea is quite limited compared to existing methods in the community.
- The baseline chosen in this paper training a not-well-adjusted model

Although the authors have addressed the concern about experiments on Imagenet-1K, the other concerns still remain, including limited novelty, compared with more recent methods, as well as more comprehensive experiments. Thus, this manuscript currently cannot meet the requirement of ICLR and the AC recommends rejection.

**Justification For Why Not Higher Score:**

Several critical issues raised by the reviewers, including limited novelty, compared with recent methods, and more comprehensive experiments. However, the authors fail to solve them in the rebuttal.

**Justification For Why Not Lower Score:**

N/A

---

### Decision · Program_Chairs · 2024-01-16

Reject